# Screening of Nutritionally Important Components in Standard and Ancient Cereals

**DOI:** 10.3390/foods13244116

**Published:** 2024-12-19

**Authors:** Vesna Dragičević, Milena Simić, Vesna Kandić Raftery, Jelena Vukadinović, Margarita Dodevska, Sanja Đurović, Milan Brankov

**Affiliations:** 1Maize Research Institute “Zemun Polje”, Slobodana Bajića 1, 11185 Zemun Polje, Serbia; smilena@mrizp.rs (M.S.); vkandic@mrizp.rs (V.K.R.); jmesarovic@mrizp.rs (J.V.); mbrankov@mrizp.rs (M.B.); 2Institute of Public Health of Serbia “Dr. Milan Jovanović Batut”, Dr Subotića Starijeg 5, 11000 Belgrade, Serbia; margarita_dodevska@batut.org.rs; 3Institute for Plant Protection and Environment, Teodora Drajzera 9, 11040 Belgrade, Serbia; stojakovicsm@yahoo.com

**Keywords:** cereals, mineral elements, antioxidants, phytic acid, phenolics, dietary fiber

## Abstract

Sustainable nutrition and food production involve dietary habits and farming systems which are eco-friendly, created to provide highly nutritious staple crops which could serve as a functional food at the same time. This research sought to provide a comprehensive analysis of whole-grain cereals, and some ancient grains toward important macro- (protein), micro-nutrients (mineral elements), and bioactive compounds, such as dietary fiber (arabinoxylan and β-glucan) and antioxidants (phytic acid, total glutathione, yellow pigment, and phenolic compounds) to provide functionality in a sustainable diet. Genotypes, such as durum wheat, triticale, spelt, emmer wheat, and barley, could be considered important and sustainable sources of protein (ranging 11.10–15.00%), as well as prebiotic fiber (β-glucan and arabinoxylan, ranging 0.11–4.59% and 0.51–6.47%, respectively), essential elements, and various antioxidants. Ancient grains can be considered as a source of highly available essential elements. Special attention should be given to the Cimmyt spelt 1, which is high in yellow pigment (5.01 μg·g^−1^) and has a capacity to reduce DPPH radicals (186.2 µmol TE·g^−1^), particularly Zn (70.25 mg·kg^−1^). The presence of phenolics, dihydro-*p*-coumaric acid, naringin, quercetin, epicatechin in grains of oats (Sopot), as well as catechin in barley grains (Apolon and Osvit) underline their unique chemical profile, making them a desirable genetic pool for breeding genotypes. This research provides a comprehensive assessment of different nutritional aspects of various cereals (some of which are commonly used, while the others are rarely used in diet), indicating their importance as nutraceuticals. It also provides a genetic background that could be translated the genotypes with even more profound effects on human health.

## 1. Introduction

Sustainable nutrition and sustainable agriculture imply eco-friendly dietary habits and farming systems, created to provide highly nutritious staple crops that will fulfill requirements for essential macro- and micro-nutrients and could be used as functional food at the same time. Modern genotypes, designed to have high yields, largely failed to meet the nutritional demand of humans and animals [1], due to the poor soil quality and climate changes. Thus, sustainable solutions are necessary. Irrespective of the increasing market demand and research on functional products, which are not only able to meet basic requirements, but also have a whole range of health benefits [2,3,4], crop diversity and gene pool diversity are able to provide reliable solutions for producing nutrient–dense food [5,6].

According to the FAO Statistical Yearbook [7], cereals are the leading crops produced worldwide (with 3.1 billion tons produced in 2022), whereas the production level of wheat increased to38% in 2022 (compared to 808 million tons in 2000) and barley production reached 155 million tons. Whole-grain cereals provide many important nutrients which may reduce the risk of developing various chronic diseases and age-related conditions, like obesity, cardiovascular diseases, metabolic syndrome, type II diabetes, and cancer [8,9,10]. From this standpoint, much more attention is being drawn to ancient wheat genotypes, such as spelt, emmer wheat, and khorasan wheat, which are relatively high in protein compared to modern wheat genotypes [9,10,11,12]. Regardless of the lower yields (compared to the modern wheat genotypes), their sustainability is one of the most important qualities, as they are highly adaptable to various environments, particularly to low-input and organic agriculture, growing on marginal soils and in dry conditions. Accordingly, old genotypes and cereals with colored grains are also able to better exploit environmental conditions and accumulate greater amounts of phytonutrients in grains [13,14,15].

To target the nutraceutical properties of cereals, it is important to identify health promoting traits, on the one hand, and antinutrients, such as phytic acid, on the other hand. Phytic acid can bind mineral elements, protein, and starch, compromising their bio-availability, so the ratio between phytic acid and mineral elements is the simplest and best measure of potential bioavailability [16,17]. However, phytic acid also exerts dietary benefits through antioxidant activity, thereby preventing lipid peroxidation [18], reducing the risk of certain cancers, cardio-vascular diseases, renal stones, etc. Its concentration in grains depends mostly on genotype, as well as on its interaction with the environment and growing conditions [14,15,19]. Lowering the concentration of phytic acid and simultaneously increasing the concentration of essential elements, such as Ca, Mg, Fe, Zn, and Mn, in grains of staple crops are goals of the bio-fortification strategy, aimed to combat malnutrition globally [20]. Various approaches, such as transgenic, conventional breeding, and agronomical approaches, are used to achieve this goal, increasing the concentration of targeted minerals and vitamins in edible parts of plants. The main focus of bio-fortification is on staple crops like rice, wheat, maize, sorghum, lupin, common bean, potato, sweet potato, and tomato. The exploitation of various fertilizers and substances that promote the absorption and accumulation of essential elements in crops, including the selection and development of desirable genotypes that can accumulate high levels of mineral nutrients are important parts of the strategy.

Sustainable plant-based protein sources are of particular importance today. While legumes are considered a main plant protein source, cereals as staple crops are targeted to elevate protein concentration in their grains too. Therefore, genetic tools should be applied to provide a more sustainable protein supply from cereal grains [21]. Moreover, whole-grain cereals are an important source of soluble and insoluble functional dietary fiber (e.g., cellulose, arabinoxylan, β-glucan, xyloglucan, and fructan), with sound health benefits, including cardio-protective association [22,23,24]. Despite this, the general consumption of dietary fiber from grains is considerably lower than the recommended levels worldwide. The most prominent dietary fibers with health benefits, as well as those that play a prebiotic role, are arabinoxylan and β-glucan [25,26]. For this reason, different breeding programs have been developed that have included research into genetic pools and translated desirable traits into improved cereal genotypes [2,27].

Bearing in mind the health-promoting bioactive compounds, whole-grain cereals are a rich source of various antioxidants in the human diet [8] and can inhibit Maillard reaction products [28]. Among cereals, ancient grains and modern genotypes with colored grains are the richest sources of phenolic compounds [11,12,29]. Their concentration and composition depend mainly on genotype and agroecological conditions. While ferulic and *p*-coumaric acids are the most common phenolics present in cereal grains [10,11,12,29], many others compounds, such as flavonoids, isoflavones, flavanones, and flavanols (catechin and epicatechin), could also be found in the cereal grains. Phenolic compounds from cereal grains also have a prebiotic role, supporting the development of beneficial microorganisms in the human gut [30,31,32].

Thus far, according to the literature review, most of the research on the nutritional quality of cereal grains is based on the examination of certain classes of compounds such as macronutrients, mineral elements, antioxidants, vitamins, etc. To our knowledge, this study is the first report on the concentration of thirty-three simultaneously evaluated parameters in whole-grain cereals, including some ancient grains, such as spelt and emmer wheat. Therefore, this study sought to evaluate the content of important macro- (protein) and micro-nutrients (mineral elements), as well as bioactive compounds, such as dietary fiber (arabinoxylan and β-glucan) and antioxidants (phytic acid, total glutathione, yellow pigment, and the specific profile of phenolic compounds) in various cereals. Furthermore, to determine the possible unique nutritive fingerprint of tested whole-grain cereals, principal component analysis was applied. This article provides extensive research on modern genotypes and ancient grains about their nutraceuticals properties, also giving essential information for breeders on the development of genotypes with enhanced grain composition.

## 2. Materials and Methods

### 2.1. Grain Material and Sample Preparation

Twenty genotypes of cereals (durum wheat, bread wheat, barley, emmer wheat, spelt wheat, triticale, rye, and oats) from the Gene Bank of the Maize Research Institute “Zemun Polje” (MRIZP), including some standard and ancient cereals, were chosen to evaluate their chemical composition from anutritional viewpoint. Grains from all genotypes which were harvested in 2022 are characterized in Appendix A.

The experiment was conducted at the experimental field of Maize Research Institute “Zemun Polje”, Belgrade (44°52′ N, 20°19′ E, 81 m a.s.l.), Serbia, on a slightly calcareous Chernozem. Texture is silt clay loam, containing the following: 51.0% sand, 31.0% silt, 18.0% clay, 3.5% organic matter, 7.0 pH KCl, and 7.5 pH H_2_O. Mineral composition revealed 68.25 mg·kg^−1^ of available N, 65.00 mg·kg^−1^ of available P, 105.85 mg·kg^−1^ of K, 264.00 mg·kg^−1^ of Mg, 2854.25 mg·kg^−1^ of Ca, 10.28 mg·kg^−1^ of Fe, 111.50 mg·kg^−1^ of Mn, 6.23 mg·kg^−1^ of Cu, and 3.46 mg·kg^−1^ of Zn in the 0-to-30 cm soil layer. The preceding crop was maize (*Zea mays* L.). Accordingly to the soil analysis, fertilization included incorporation of the 250 kg urea ha^−1^ in the spring of 2022.

Grain samples (approximately 0.5 kg) were dried in a ventilation oven at 60 °C (EU instruments, EUGE425, Novo Mesto, Slovenia, EU) to obtain uniform moisture content (12%) and then milled on Perten 120—Hägersten, Sweden (particle size < 500 μm). 

### 2.2. Chemicals and Reagents

Gallic acid (GA), catechin (CAT), dihydrocaffeic acid (diCA), epicatechin (EPI), dihydro-p-coumaric acid (di-p-CoumA), naringin (NA), daidzein (DA), cinnamic acid (CIN), naringenin (NG), chlorogenic acid (ChlA), caffeic acid (CA), p-coumaric acid (p-CoumA), ferulic acid (FA), isoferulic acid (IFA), quercetin (QUE), as well as acetone, formic acid, methanol, all HPLC-grade, were obtained from Sigma-Aldrich (Munich, Germany). From the same supplier, Folin–Ciocalteu reagent, butanol, sodium acetate, sodium carbonate, and trichloroacetic acid (TCA) were also purchased. TRIS (2-amino-2-hydroxymethyl-propane-1,3-diol), DTNB (2,2′-dinitro-5,5′-dithiobenzoic acid), iron (III) chloride hexahydrate, DPPH• (2,2-diphenyl-1-picrylhydrazyl), 5′-sulfosalicylic acid, TROLOX (6-hydroxy-2,5,7,8-tetramethylchroman-2-carboxylic acid), and TPTZ (2,4,6-Tris(2-pyridyl)-s-triazine) were purchased from Merck (Darmstadt, Germany). Ammoniumheptamolybdate tetrahydrate, ammonium metavanadate hydrochloric acid, and nitric acid were purified. Ultrapure water, with a conductivity of 0.055 μS/cm (TKA, Niederelbert, Germany), was used to prepare the appropriate standard solutions and samples, while Target2^®^ syringe filters (PTF membrane, 0.45 μm, diameterof17 mm) were used to filter the extracts. Enzymatic assay kits K-XYLOSE and K-BGLU were purchased from Megazyme (Bray, Ireland).

### 2.3. Determination of Protein Content

The protein content from whole grains was determined using an infrared analyzer (Infraneo, Chopin Technologies, Villeneuve-la-Garenne Cedex, France). For the analysis, a sample of approximately 0.5 kg was passed through the sliding cell and the near-infrared spectra range was 700 nm to 1100 nm using a high-resolution monochromator (0.1 nm). The protein content was reported as the mean value of two replications and expressed in g per 100 g (%) of the sample.

### 2.4. Determination of the Concentration of Phytic Phosphorus, Inorganic Phosphorus, and Total Glutathione

Phytic acid (Pphy), inorganic phosphorus (Pi), and total glutathione (GSH) from the analyzed samples (approximately 0.5 g) were extracted with the 5% TCA solution (10 mL) using avertical shaker (30 min, at room temperature). After centrifugation (12,000 rpm for 15 min (Dynamica—Model Velocity 18R Versatile Centrifuge, Rotor TA15-24-2), at 4 °C, the obtained clear supernatant was used for further analysis.

#### 2.4.1. Determination of Phytic Acid and Inorganic Phosphorus

Pphy and Pi were determined using the method of Dragičević et al. [33]. For phytic acid determination, the distilled water (1.25 mL) and Wade reagent containing the 0.3 g of FeCl_3_ × 6H_2_O and 3 g of 5′-sulfosalicylic acid in one liter (0.5 mL) were added to previously obtained supernatant (0.25 mL). After developing the pink color, absorbance was measured using the spectrophotometer (Biochrom Libra S22 UV/Vis Spectrophotometer—Biochrom, UK) at λ = 500 nm against a blank probe (containing 5% TCA). For Pi determination, an aliquot of previously obtained supernatant (1 mL) was mixed with distilled water (1 mL) and Pi reagent (0.5 mL). The Pi reagent consisted of 6% ammoniumheptamolybdate (500 mL) and 0.25% ammonium metavanadate mixed with 4.74 M nitric acid (500 mL). Absorbance was measured using a spectrophotometer at λ = 400 nm against a blank probe (containing 5% TCA). The Pphy concentration and the Pi concentration were reported as the mean values of three replications and expressed in mg per g of the DW sample.

#### 2.4.2. Determination of Total Glutathione

To determine total glutathione (GSH) [34], an aliquot of previously obtained supernatant (0.5 mL) was mixed with distilled water (0.5 mL) and reagent consisting of 0.1 M TRIS (pH = 7) and 1.5 mM of DTNB (1 mL). After centrifugation, the absorbance of the extract was measured at 412 nm against a blank probe (containing 5% TCA). The GSH concentration was calculated based on the measured absorbance values of the samples and the molar absorption coefficient of 1.36 × 10^4^ dm^3^ mol^−1^ cm^−1^ at 412 nm. The GSH concentration was reported as the mean value of three replications and expressed in nmol per g of the DW sample.

### 2.5. Determination of Total Phosphorus

Total phosphorus (P) determination was achieved using the method ISO 13730 [35] with some modifications. The analyzed samples were firstly subjected to dry incineration. Then, after homogenization with nitric acid (4.81 M), followed by the heating in a boiling water bath (30 min), the extracts were cooled and filtered. After adding the Pi reagent (used for determining inorganic phosphorus) samples were left to stand for 15 min, and then the absorbance was measured at 430 nm against a blank probe (containing nitric acid and Pi reagent) using a spectrophotometer. The P concentration was reported as the mean value of three replications and expressed in g·kg^−1^ of the DW samples.

### 2.6. Determination of Yellow Pigment

The concentration of yellow pigment (YP) was determined using the method of Vančetović et al. [36]. After extracting approximately 2 g of flour with 10 mL of saturated 1-butanol using a vertical shaker (30 min, at room temperature) and conducting centrifugation at 10,000 rpm for 5 min, the absorbance was measured at λ = 436 nm. The β-carotene (βCE) concentration was reported as the mean value of three replications and expressed in μg per g of the DW samples.

### 2.7. Identification and Determination of Total Phenolic Compounds and Antioxidant Activity

The total phenolic compounds (TPCs) were determined and the antioxidant activity was obtained according to the procedure described by Singleton et al. [37]. Approximately 2 g of flour was extracted with 10 mL of 70% methanol (MeOH) using an ultrasound bath (A-Sonic, frequency of 400 Hz) at 2 × 30 min and after centrifugation at 10,000 rpm (5 min), a clear supernatant was used for the determination of both total phenolic concentration (TPC) and antioxidant activity. An aliquot of 0.2 mL was mixed with 0.5 mL of distilled water. Then, 0.25 mL of Folin–Ciocalteu reagent (2 M relative to the acid) and 1.25 mL of Na_2_CO_3_ (20%) were added and the cuvettes were vigorously shaken and left in the dark for 40 min. After that, the absorbance was measured at 722 nm against a blank probe (containing MeOH). The concentration of the TPCs was reported as the mean value of four replications and expressed in mg of gallic acid equivalents (GAEs) per g of the DW sample.

The total phenolic compounds (TPCs) were determined and the antioxidant activity was obtained according to the procedure described by Singleton et al. [37]. Approximately 2 g of flour was extracted with 10 mL of 70% methanol (MeOH) using an ultrasound bath (A-Sonic, frequency of 400 Hz) at 2 × 30 min and after centrifugation at 10,000 rpm (5 min), a clear supernatant was used for the determination of both total phenolic concentration (TPC) and antioxidant activity. An aliquot of 0.2 mL was mixed with 0.5 mL of distilled water. Then, 0.25 mL of Folin–Ciocalteu reagent (2 M relative to the acid) and 1.25 mL of Na_2_CO_3_ (20%) were added and the cuvettes were vigorously shaken and left in the dark for 40 min. After that, the absorbance was measured at 722 nm against a blank probe (containing MeOH). The concentration of the TPCs was reported as the mean value of four replications and expressed in mg of gallic acid equivalents (GAEs) per g of the DW sample.

DPPH radical scavenging activity was performed using the method described by Brand-Williams et al. and Wong et al. [38,39]. The obtained clear supernatant was diluted 1:100 in MeOH prior to the analysis and 200 μL of aliquots was mixed with 3.8 mL of DPPH• reagent (0.1 mM). After incubation (30 min in dark, 25 °C), the absorbance was measured at 517 nm against a blank probe (containing MeOH). The DPPH radical scavenging activity was expressed in μmol of TROLOX equivalents (TEs) per g of the DW sample and was reported as the mean value of four replications.

Antioxidant activity was also determined using the ferric-reducing antioxidant power assay (FRAP) method [39,40]. The obtained clear supernatant (0.15 mL) was mixed with 4.5 mL of FRAP reagent (2.5 mL of 10 mM TPTZ was dissolved in 40 mM HCl and then mixed with a 2.5 mL solution of 20 mM FeCl_3_ × 6 H_2_O in acetate buffer pH = 3.6). After the incubation (30 min, 37 °C), absorbance was measured at 593 nm against a blank probe (0.15 mL MeOH + 4.5 mL FRAP reagent). The antioxidant activity was expressed in μmol Fe^2+^ per g of DW sample and it was reported as the mean value of four replications.

### 2.8. Determination of Arabinoxylan and β-Glucan

Arabinoxylan was determined by a spectrophotometric method using the enzymatic assay kit K-xylose (Megazyme, Bray, Ireland). The assay was performed according to the instruction manual of the kit producer. Generally, the conversion of α-D-xylose to β-D-xylose is catalyzed by xylose mutarotase. The obtained β-D-xylose is then oxidized by NAD+ to D-xylonic acid in the presence of β-xylose dehydrogenase (pH 7.5). The obtained amount of NADH formed in this reaction is proportional to the D-xylose concentration and it is measured by the increase in absorbance at 340 nm. The concentration of arabinoxylan was calculated according to Formula (1):(1)$$Arabinoxylan=Concentration\;of\;D-xy\;lose\;(g/100\;g)\times \frac{100}{62}\;(g/100\;g)$$

β-glucan was quantified by a spectrophotometric method [41] using the enzymatic assay kit K-BGLU (Megazyme, Bray, Ireland). The samples were suspended and hydrated in a buffer solution (pH 6.5) and then incubated with purified lichenase enzyme and filtered. Subsequently, an aliquot of the filtrate was hydrolyzed to completion with purified β-glucosidase. The D-glucose produced was assayed using a glucose oxidase/peroxidase reagent and the absorbance was measured at 510 nm.

### 2.9. HPLC Analysis

The identification and quantification of phenolic compounds (gallic acid (GA), catechin (CAT), dihydrocaffeic acid—(DiCA), epicatechin (EPI), dihydro-p-coumaric acid (di-p-CoumA), naringin (NA), daidzein (DA), cinnamic acid (CIN), naringenin (NG), chlorogenic acid (ChlA), caffeic acid (CA), p-coumaric acid (p-CoumA), ferulic acid (FA), isoferulic acid (IFA), and quercetin (QUE)) were determined in flour using high-performance liquid chromatography (HPLC system Shimadzu Nexera XR) equipped with column Zorbax SB C18 (4.6 × 250 mm, pore diameter 5 μm). The mobile phase A was 0.1% HCOOH in H_2_O, while the mobile phase B was 0.1% HCOOH in MeOH. The used linear gradient was adopted [42], starting from 95% A to 70% A after 25 min, 60% A after 35 min, 52% A after 40 min, 30% A after 50 min, 0% A after 55 min, 95% A after 65 min, and 95% A after 75 min. The flow rate was 1 mL min^−1^, with 10 µL of the injected sample being used. The column temperature was set at 25 °C and the wavelengths were 254 nm, 280 nm, and 325 nm. The LabSolutions Shimadzu program was used for instrument control, as well as for data acquisition and analysis. Calibration curves were used for the quantitative calculation of each phenolic compound. The concentration was reported as the mean value of three replications and expressed in μg per g of the DW sample.

### 2.10. Mineral Element Analysis

The concentration of potassium (K), sodium (Na), magnesium (Mg), calcium (Ca), copper (Cu), iron (Fe), manganese (Mn), selenium (Se), zinc (Zn), and nickel (Ni) in flour of samples was determined by plasma inductively spectrometry coupled with optical emission (ICP-OES) using a ICP 7400 Duo Thermo Scientific, UV/VIS 214–766 nm. Samples were prepared after dry incineration, except for Se where the acid hydrolysis process was applied, according to the procedure described by the Association of Official Analytical Chemists [43]. Quantifying mineral composition was performed based on Multi–Element Plasma Standard Solution 4 (Thermo Scientific, Berlin, Germany) and the results were expressed ing·kg^−1^ of dry grain samples for K, as well as mg·kg^−1^ for Na, Mg, Ca, Cu, Fe, Mn, Se, Zn, and Ni. The Se concentration was under the limit of the detection, while trace amounts of Ni were only determined in three samples of oats, so they were not included in the results.

### 2.11. Statistical Analysis

To assess the data on tested biochemical traits, one-factorial analysis of variance (ANOVA) was applied using the SPSS for Windows Version 15.0 (SPSS, 2006). In order to determine the differences between the samples, the Tukey HSD test was performed at a 0.95 confidence level (*p* ≤ 0.05). Moreover, MATLAB (R2011a) with PLS Toolbox software package (v.6.2.1) was used for principal component analysis (PCA) and agglomerative hierarchical cluster analysis (HCA). For these analyses, the obtained data were mean-centered and auto-scaled, and the singular value decomposition (SVD) algorithm was employed (at a 95% confidence level) for Hotelling T2 limits.

## 3. Results

### 3.1. The Concentration of Protein, Antioxidants, and Dietary Fiber in the Standard and Ancient Grains

Examined genotypes expressed the significant variability in the concentration of the following important nutrients: protein, Pi, Pphy, YP, TPCs, GSH, β-glucan, and arabinoxylan (Table 1). Greater differences in protein concentration were between durum wheat genotypes, where Dur1 had the greatest concentration (15%). The lowest concentration was in spelt wheat (9.85%). Bar2 had the highest concentration of Pi, while Bar1 was the highest in GSH and β-glucan, and Bar2 had the lowest Pi and highest β-glucan. Among all tested samples, the greatest concentration of Pphy and YP was noticed in spelt grain, while the lowest Pphy value was found in Oats3 and the lowest concentration of YP was in Trit1. Oat samples Oats2 and Oats1 showed a high TPC and arabinoxylan concentration, respectively. Contrary, the lowest TPC concentration was found in BW1 and the lowest arabinoxylan concentration was noticed in the sample of spelt.

### 3.2. Essential Elements and Their Ratios with Phytic Acid in the Standard and Ancient Grains

The significant differences among tested genotypes in the accumulation of essential elements (P, K, Ca, Mg, Na, Fe, Mn, Zn, and Cu) (Table 2) could also be ascertained. Among the tested essential elements, the greatest concentration of Cu was recorded in grains of Em2, while the lowest value was found in the grains of Bar2. The spelt grain had the highest concentration of P, K, Mg, Mn, and Zn. The lowest concentration of P was found in BW1, while the lowest concentration of Mn and Zn was recorded in the Bar3 grain. The lowest concentration of K, Mg, and Ca was noticed in the grain of Em2, whereas the highest Ca concentration was found in the Oats1 grain. The highest concentration of Na and the lowest Fe concentration were noticed intheTrit2 grain. Moreover, the lowest concentration of Na was found in Oats2, and the highest Fe concentration was observed in the Rye1 grain.

According to the cluster analysis encompassing all examined elements, the following clusters were present (Figure 1). The first consisted of only Oats2 and the second one was Em2. The third comprised all other genotypes, consisting of two sub-clusters (the first one consisted of Oats1 and Oats3, while the second one consisted of all the other genotypes).

Among all tested genotypes, the lowest ratio between phytic acid and essential elements (Phy/Ca, Phy/Mg, Phy/Fe, and Phy/Mn) was in Oats3, while the lowest Phy/Zn was in spelt (Appendix A). The greatest Phy/Ca and Phy/Mg values were in Em2 grain; the highest Phy/Mn and Phy/Zn values were in Bar3, whereas the greatest Phy/Fe value was in the Trit2 grain. According to cluster analysis (Figure 2), two clusters were present, with spelt independent of other genotypes that formed three sub-clusters. Regarding the first sub-cluster, samples of Dur1, Dur2, and Em2 formed the first subgroup; the samples of Dur3, Bar3, Rye2, and Bar4 created the second subgroup; and the Em1, Oats2, and Oats3 were grouped in the third subgroup. The second sub-cluster consisted of BW1, BW2, Trit1, Trit2, Trit3, and Oats1, while the samples of Bar1 and Rye1 established the third sub-cluster.

### 3.3. Phenolic Acids and Antioxidant Status in the Standard and Ancient Grains

There was great variability in the concentration of certain phenolic compounds among the tested genotypes, while some of them had concentrations under the limit of detection. Oat grains were observed to have the highest concentration in the majority of examined phenolic compounds (Table 3). According to the obtained results, Oats1 had the highest concentration of EPI, NA, DA, and QE; Oats2 had the highest concentration of PA, CA, and FA; andOats3 had the greatest concentration of p-CoumA. The highest concentration of NG was found in Em2. The highest concentration of diCA, IFA, and CIN was observed in samples Trit2, Trit3, and BW1, respectively. Barley genotypes were also relatively high in phenolic compounds, where Bar2 had the greatest concentration of GA and CAT and Bar4 had the greatest concentration of ChlA.

When the antioxidant activity was considered, the greatest value of capacity to reduce DPPH radical was noticed for durum wheat genotypes, particularly for Dur1, although the greatest FRAP values were obtained in oat samples, especially in Oats2 (Table 4).

According to the cluster analysis, which encompassed all examined antioxidants, including the capacity to reduce DPPH radicals, as well as FRAP, two clusters were formed (Figure 3). The samples Trit1 and Oats3 constituted the first cluster among all tested samples. The second cluster encompassed all other genotypes, which were divided into two sub-clusters. The first one consisted of samples Dur1 and Dur2. Under the second sub-cluster, two sub-sub-clusters were formed, encompassing Dur3, Em2, all barley genotypes, both rye genotypes, as well as Oats1 and Oats2 in the first sub-sub-cluster. The second sub-sub-cluster included both bread wheat genotypes Trit2, Trit3, spelt, and Em1.

### 3.4. PC Analysis

To access and evaluate the possible linkage between tested genotypes and phytochemicals present in all examined genotypes, PCA was performed. PC analysis showed a six-component model which explained 87.51% of the total variability. The first axis had a total variability of 34.1%, while the second axis had 18.9%, the third axis had 12.6%, and the fourth axis had 10.7% of total variability. PCA showed that the concentration of arabinoxylan, Ca, FA, and QUE correlated with the first axis in a significantly positive way, while Pphy correlated with the first axis a in significantly negative way. Zn and p-CoumA correlated with the second axis in a significantly positive way. Pi, GSH, βGlu, and P correlated with the third axis in a significantly positive way, while Mn and NA correlated with the same axis in a significantly negative way; Fe and Cu correlated with the fourth axis in a significantly positive way, while K and Mg correlated with the same axis in a significantly negative way (Appendix A). The best separation after PCA resulted in the model consisting of the first two axes (Figure 4). Results revealed that all three oat genotypes were mostly determined by the concentration of arabinoxylan, NA, and QUE. The best separation after PCA resulted in the model consisting of the first two axes (Figure 4). Results revealed that all three oat genotypes were mostly determined by the concentration of arabinoxylan, NA, and QUE, while the Bar1, Bar2, and Bar3 genotypes, together with Trit2, were found to be associated with protein content. The Bar4 genotype was associated with the Pi concentration. To a lesser extent, barley was found to be associated with B-gluc and GSH. According to the PCA, the oat and barley genotypes, with the exception of Bar4, were separated from all other species.

## 4. Discussion

Cereals are an important staple food worldwide, providing most of the macro- and micro-nutrients to humans. Owing to their high starch concentration, they are the foremost source of energy in the diet, but other nutrients, with functional properties, make them nutritious and desirable parts of any diet. From anutritional viewpoint, the diversity of species (wheat, rye, barley, oats, etc.) and species’ genotypes could offer a whole range of bioactive compounds in nutrition. It is well known that most of the present-day genotypes are designed for high yields with high starch content; however, they are low in essential elements and bioactive compounds and poorly adaptable to stressful conditions. Thus, ancient and some older and colored grains are attracting attention nowadays as they are highly adaptable and tolerant to growth in limited conditions [13,14,44]. Importantly, they are rich in protein and various bioactive compounds, so they can therefore be considered nutraceuticals [9].

### 4.1. Concentration of Protein and Dietary Fiber in the Standard and Ancient Grains

The results showed that protein in samples of Bar2, Bar4, Trit3, Dur1, and spelt had concentrations ≥ 13%, indicating significant variability among analyzed genotypes in the phytochemical composition of the grains. Majzoobi et al. [10] obtained a wide range of protein content among various grains, including barley, spelt, and emmer wheat. They also accentuated that spelt and barley are the richest sources of dietary fiber, too. Our research revealed that the concentration of β-glucan, as a soluble fiber [4,45], was the highest in the samples of barley (particularly Bar2) and oat genotypes, while the samples of oats (mainly Oats1) and rye genotypes were found to be the best sources of arabinoxylan [2,45]. High concentrations of β-glucan and arabinoxylan in examined genotypes of barley, rye, and particularly oats can contribute to the beneficial microbiota in the human gut [46], and they can also be a fiber source for gluten-free bread production [47]. More importantly, examined triticale and barley genotypes could present a gene pool for the development of high-protein cereals.

### 4.2. Essential Element Levels in the Standard and Ancient Grains

Considering essential elements, examined genotypes of rye, oats, emmer, and spelt wheat had generally higher concentrations of essential elements compared to other genotypes. In contrast to barley genotypes (especially Bar2) with the highest Pi concentration, triticale and spelt revealed the highest concentration of Pphy, which is the most important factor that sequesters most of the essential elements [16,17]. Nevertheless, spelt was also the richest source of K, Mg, Mn, and Zn, which might suggest that it is capable of absorbing and accumulating higher amounts of minerals from the soil, making it a superior crop for growth in low-input systems [15]. Together with a high concentration of YP, the tested spelt genotype can be seen as a highly nutritious grain as it is unique and independent of other genotypes in terms of the potential bioavailability of essential elements. Similarities between spelt, durum wheat, and triticale genotypes in terms of the accumulation of essential elements indicate their increased ability to utilize environmental sources concerning the other cereals. The tested genotypes of emmer wheat and barley and, to a lesser extent, durum wheat are also capable of accumulating higher amounts of Fe, Mn, and Cu, while oats could be considered a good Ca source.

However, the lowest Pphy concentration refers to the higher potential availability of essential elements from the Oats3 grain, which ensures the enhanced absorption of Ca, Mg, Fe, and Mn during the digestion process [48,49]. A lower Phy/essential elements ratio was also present in rye and emmer wheat genotypes, but to a lesser degree.

### 4.3. Antioxidants and Phenolic Acids in the Standard and Ancient Grains

Plants synthesize antioxidants to combat oxidative stress, and, in seeds, they serve to protect viability during storage [50,51]. The concentration and antioxidant potential of phenolic compounds are important traits from a nutritional point of view, too [52]. In this research, some critical low-molecular-weight non-enzymatic antioxidants were analyzed in the grain of examined cereal genotypes, including the concentration of YP as a fat-soluble antioxidant, as well as the concentration of Pphy, TPC, and GSH as water-soluble antioxidants.

The antioxidant capacity to reduce DPPH radicals was shown to be the highest in Dur1, which was also a genotype with the highest GSH concentration. High DPPH radical scavenging activity was also noted in spelt (which had the greatest concentration of Pphy and YP), as well as Em2 and Rye2 (which were high in TPCs and GSH). The FRAP value was unsurprisingly the highest in Oats2, which showed the highest TPC value among all the tested samples. The increased variability of Pphy, YP, and GSH, along with pronounced similarity between examined durum wheat genotypes, Em2, spelt, and triticale genotypes, makes them a desirable source of antioxidants, as well as a gene pool for the further development of nutraceutical cereals.

In this research, the concentration of some common and rarely analyzed phenolic compounds in cereal grains has been shown. The results from this study revealed the presence of quercetin, daidzein, naringin, naringenin, catechin, and epicatechin in the tested whole-grain cereals. Such data represent novelty because, through literature review, it has been shown that the above-mentioned compounds can be mostly found in plant species, such as buckwheat [53], soybeans [54,55], quinoa [56], citrus fruit peels [57], apples, and Malus doumeri fruit [58,59]. Due to the lack of information on the phytochemical content in whole-grain cereals, and especially of ancient grains, such as spelt and emmer wheat, these results can be useful in enhancing data on this subject, especially nowadays when the quest for high nutritional quality in food is a world tendency.

Montevecchi et al. [11] showed in their study that ancient wheat varieties, with higher adaptability to local conditions and marginal soils, are a rich source of various phenolic compounds, containing conjugated flavones, such as vitexin and isovitexin. Due to the fact that p-coumaric acid and ferulic acid are the most present phenolics in cereal grains [29], they were observable in the grains of all examined genotypes with the highest concentration achieved in Oats3 and Oats2. While the greatest variability in the concentration of phenolic compounds was present in the grains of barley and oats [29,60], some specificity was present, especially when flavonoids were considered. Thus, samples of Oats1 and Oats2 showed the highest variability in naringin, quercetin, epicatechin, dihydro-*p*-coumaric, ferulic, and isoferulic acid, while the Trit2 samples showed the highest variability in dihydrocaffeic acid. The naringenin was mainly found in the grains of Em2 and oats. Since observed phenolic acids have profound health benefits [61], this research supports the statement that whole-grain products, particularly examined oat samples, are a valuable source of various micro- and trace-bioactives. Owing to the high level of dietary and prebiotic fiber [25] and prebiotic phenolic compounds [30], examined oat genotypes could be considered nutraceuticals.

### 4.4. PC Analysis

PCA was conducted to uncover the potential unique nutritional fingerprint of the tested whole-grain cereals. PCA revealed that the oats and barley genotypes were separated from all other analyzed species owing to their unique nutritive profile, while all the other species were similar due to their phylogenetic relationship. The association of oat genotypes with arabinoxylan, QUE, and NA, along with Ca, TPCs, FA, and pCoumA, to a minor extent, highlights their nutritional importance [10]. These compounds contribute significantly to antioxidant activity, play a prebiotic role [30], and serve as an important source of calcium. As previously mentioned, a possible association between the Bar1, Bar2, and Bar3 samples and the concentrations of protein, β-Glu, and GSH suggests the distinctiveness of barley genotypes as a valuable source of important bioactive compounds. The results obtained in our study are in agreement with those obtained in the study of Majzoobi et al. [10] who characterized barley as a highly nutritious grain, abundant in protein, vitamins, and minerals. These authors also emphasized the importance of barley’s high β-glucan content, along with its phytosterols and polyphenols. Additionally, the strong association between triticale genotypes and protein content further supports the idea that triticale is a highly desirable and valuable nutritious crop. Consistent with our findings, Camerlengo and Kiszonas [62] recognized triticale as a nutrient-rich crop, offering higher levels of protein, health-promoting compounds, and a well-balanced amino acid composition. These findings make these cereals highly desirable and offer breeders the potential to develop genotypes with enhanced nutritional value.

## 5. Conclusions

While the importance of whole-grain cereals as a source of various nutrients has been confirmed, the comprehensive analysis conducted in this study shed new light on their nutritional uniqueness and importance as staple food and nutraceuticals at the same time. Some of the genotypes that were analyzed, including durum wheat, triticale, spelt, emmer wheat, and barley, emerge as significant and sustainable sources of protein. Additionally, their content of bioactive compounds—such as prebiotic fibers (β-glucan and arabinoxylan), essential minerals, and antioxidants—further enhances their value, positioning them as valuable ingredients for functional foods.

Owing to the high protein, and particularly GSH level, Cosmostar (durum wheat variety) could present a genetic pool for the development of high protein/GSH genotypes. While ancient grains (emmer and spelt wheat) are highly adaptable and sustainable crops, according to this study, Emmer FON, Emmer LP2-1-5, and Cimmyt spelt 1 are important sources of highly available essential elements. Special attention should be given to Cimmyt spelt 1, a genotype high in yellow pigment, which possesses high concentration of essential elements (particularly Zn) and a high capacity to reduce DPPH radicals. Thus, ancient grains from the MRIZP collection could serve as candidates for bio-fortification studies. The high concentration of phenolic compounds and the presence of phenolics, dihydro-*p*-coumaric acid, naringin, quercetin, epicatechin in grains of oats (Sopot), and catechin in barley grains (Apolon and Osvit) underline their unique nutritive profile, making them a desirable genetic pool for breeding cereal genotypes high in antioxidants and prebiotic compounds.

This study is the first report of the assessment the concentration of thirty-three parameters simultaneously in whole-grain cereals, including some ancient grains, such as spelt and emmer wheat. For the first time, the PC analysis revealed a distinctive nutritional profile for barley and oat samples. These findings emphasize the potential of these cereals as a source of important nutrients and phytochemicals, offering the possibility of developing genotypes with improved health benefits for humans.

## Figures and Tables

**Figure 1 foods-13-04116-f001:**
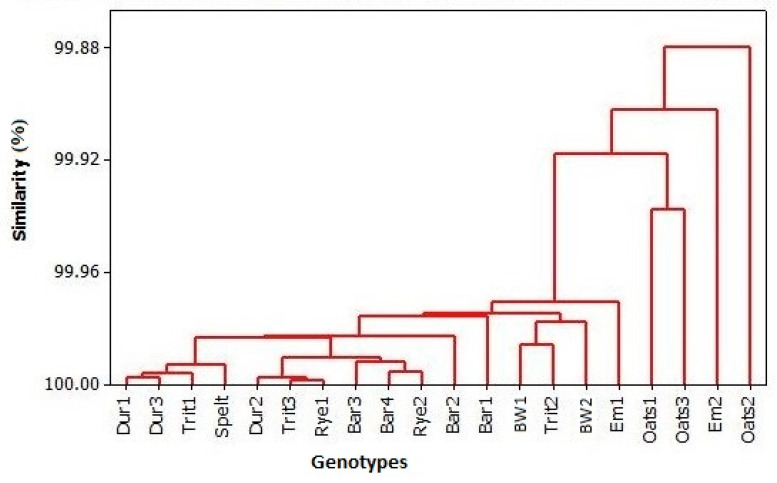
Dendrogram presenting the concentration of mineral elements in standard and ancient grains.

**Figure 2 foods-13-04116-f002:**
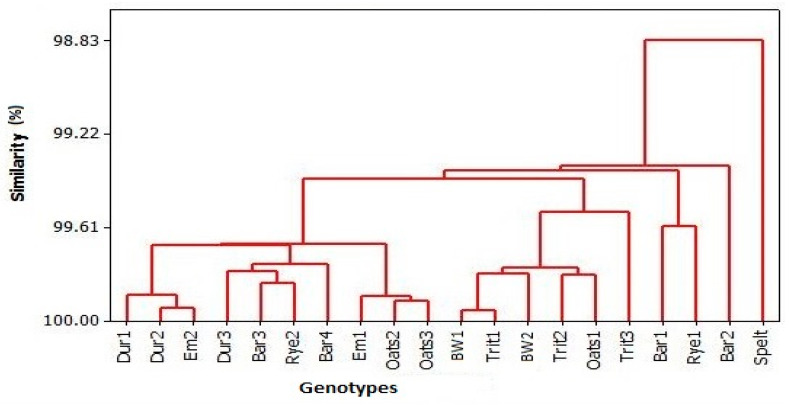
Dendrogram presenting ratio of Phy and essential elements in standardgrains and ancient grains.

**Figure 3 foods-13-04116-f003:**
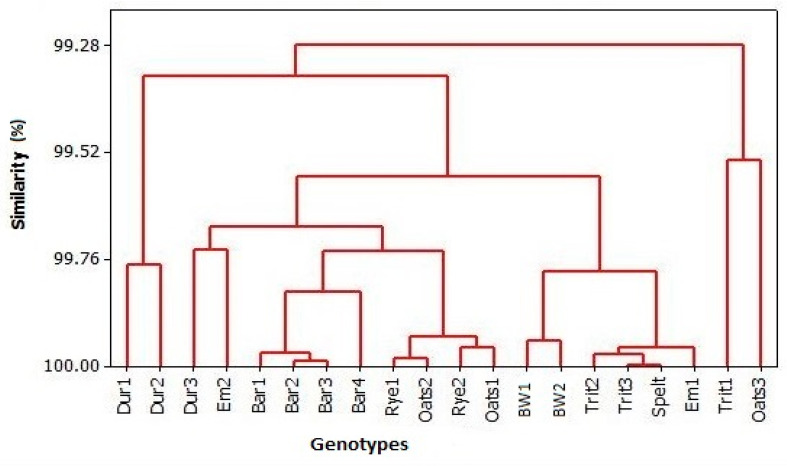
Dendrogram presenting concentrations of antioxidants in the grain of examined standard and ancient grains.

**Figure 4 foods-13-04116-f004:**
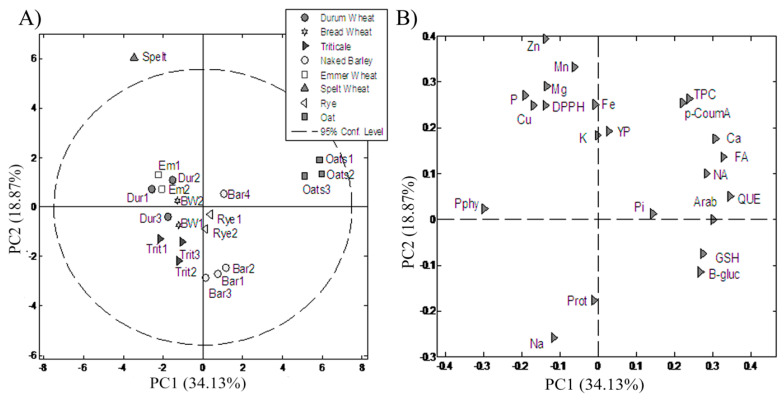
PCA for evaluated phytochemicals in tested genotypes: (**A**) score and (**B**) loading plot. Abbreviations: protein (Prot), inorganic phosphorus (Pi), phytic phosphorus (Pphy), yellow pigment (YP), total phenolic compounds (TPCs), total glutathione(GSH), β-glucan (β-glu), arabinoxylan (Arab), phosphorus (P), potassium (K), calcium (Ca), magnesium (Mg), sodium (Na), iron (Fe), manganese (Mn), zinc (Zn), copper (Cu), naringin (NA),p-coumaric acid(p-CoumA), ferulic acid (FA), and quercetin (QUE).

**Table 1 foods-13-04116-t001:** The concentration of protein, inorganic phosphorus (Pi), phytic phosphorus (Pphy), yellow pigment (YP), total phenolics (TPCs), total glutathione (GSH) and dietary fiber (β-glucan and arabinoxylan) in the examined standard and ancient grains.

	Protein			Pi			Pphy			YP			TPCs			GSH			β-glucan			Arabinoxylan		
	%			mg·g^−1^						μg·g^−1^						nmol·g^−1^			g 100·g^−1^					
Dur1	15.03 a	±	0.32 *	0.37 k	±	0.019	4.80 bc	±	0.02	2.75 hi	±	0.06	1402. 7 h	±	3.51	77.86 k	±	2.80	0.11 h	±	0.01	5.55 i	±	0.21
Dur2	11.93 ef	±	0.24	0.50 i	±	0.005	4.54 defg	±	0.02	4.70 b	±	0.15	1240.9 i	±	9.23	85.05 jk	±	5.99	0.13 h	±	0.02	5.39 j	±	0.18
Dur3	12.60 bcd	±	0.48	0.48 i	±	0.003	4.50 fg	±	0.06	4.39 c	±	0.04	926.0 k	±	14.69	96.86 jk	±	5.36	0.14 h	±	0.01	4.81 j	±	0.31
BW1	10.50 ij	±	0.22	0.39 jk	±	0.013	4.48 fg	±	0.00	3.40 f	±	0.09	735.4 l	±	7.00	130.57 i	±	5.99	0.17 h	±	0.02	6.19 h	±	0.26
BW2	10.53 hij	±	0.26	0.37 k	±	0.005	4.73 cde	±	0.06	3.05 g	±	0.10	1550.1 f	±	1.48	107.41 ij	±	9.98	0.19 h	±	0.04	6.74 f	±	0.35
Trit1	12.53 cde	±	0.26	0.41 j	±	0.012	5.02 ab	±	0.06	2.54 j	±	0.06	789.0 l	±	8.90	226.82 f	±	2.38	0.18 h	±	0.01	6.50 fg	±	0.30
Trit2	12.03 de	±	0.36	0.55 h	±	0.003	4.67 cdefg	±	0.12	2.66 ij	±	0.06	1115.9 j	±	8.08	179.30 g	±	2.78	0.23 h	±	0.03	6.47 g	±	0.25
Trit3	13.00 bc	±	0.83	0.49 i	±	0.005	4.68 cdef	±	0.09	2.90 gh	±	0.02	925.8 k	±	5.45	188.87 g	±	5.99	0.13 h	±	0.01	6.32 gh	±	0.21
Bar1	11.13 g	±	0.54	0.87 b	±	0.010	4.46 g	±	0.00	3.77 e	±	0.11	1115.1 j	±	11.18	607.34 a	±	21.16	4.15 b	±	0.28	4.25 l	±	0.34
Bar2	14.50 a	±	0.24	0.91 a	±	0.015	4.50 fg	±	0.01	3.67 e	±	0.04	1784.5 d	±	6.19	399.74 d	±	4.36	4.59 a	±	0.38	3.86 m	±	0.37
Bar3	11.23 g	±	0.27	0.72 e	±	0.019	4.70 cdef	±	0.17	3.61 e	±	0.13	924.1 k	±	8.88	364.56 e	±	7.59	3.78 c	±	0.40	4.54 k	±	0.30
Bar4	13.17 b	±	0.53	0.60 g	±	0.010	4.19 h	±	0.05	4.16 d	±	0.02	2175.2 b	±	4.18	498.73 b	±	2.80	3.44 cd	±	0.18	6.26 gh	±	0.33
Em1	10.12 jk	±	0.18	0.54 h	±	0.011	4.53 efg	±	0.07	2.63 ij	±	0.15	1486.6 g	±	1.61	155.36 h	±	2.76	0.13 h	±	0.03	3.84 m	±	0.32
Em2	10.10 jk	±	0.30	0.69 ef	±	0.011	4.76 cd	±	0.04	3.65 e	±	0.02	1631.4 e	±	11.87	170.73 gh	±	3.37	0.17 h	±	0.04	4.81 j	±	0.30
Spelt	9.85 k	±	0.23	0.82 c	±	0.013	5.22 a	±	0.02	5.01 a	±	0.06	1735.5 d	±	4.66	217.64 f	±	4.37	0.39 gh	±	0.08	0.51 n	±	0.17
Rye1	11.03 ghi	±	0.61	0.72 e	±	0.005	4.13 h	±	0.01	4.19 d	±	0.15	1261.2 i	±	7.46	397.13 d	±	0.57	0.62 fg	±	0.04	7.25 e	±	0.33
Rye2	11.07 ghi	±	0.51	0.56 h	±	0.007	4.04 h	±	0.10	4.04 d	±	0.02	1082.8 j	±	11.87	370.17 e	±	8.37	0.85 f	±	0.13	7.62 d	±	0.28
Oats1	11.40 fg	±	0.32	0.63 g	±	0.008	4.06 h	±	0.09	4.73 b	±	0.10	2359.1 a	±	3.98	473.78 c	±	9.37	3.05 e	±	0.17	12.69 a	±	0.33
Oats2	11.10 gh	±	0.29	0.76 d	±	0.000	4.00 h	±	0.11	3.26 f	±	0.15	2404.6 a	±	10.25	481.18 bc	±	8.37	3.70 c	±	0.39	12.04 b	±	0.25
Oats3	11.13 g	±	0.84	0.67 f	±	0.022	3.49 i	±	0.07	2.81 hi	±	0.02	1981.7 c	±	1.18	461.39 c	±	6.94	3.31 de	±	0.48	10.65 c	±	0.73
Tukey HSD (*p* < 0.05)	0.11			0.006			0.041			0.036			9.934			4.246			0.066			0.048		

* Average ± standard error; the values signed with the same letter are not significantly different at the 0.05 level of significance.

**Table 2 foods-13-04116-t002:** Essential elements in the grain of the examined standard and ancient grains.

	P			K			Ca			Mg			Na			Fe			Mn			Zn			Cu		
	g·kg^−1^						mg·kg^−1^																				
Dur1	6.12 de	±	0.05 *	2.99 h	±	0.02	397.0 k	±	4.94	1274.0 b	±	8.82	73.50 c	±	3.45	51.65 d	±	1.76	30.40 f	±	0.98	38.20 b	±	1.63	7.10 b	±	0.24
Dur2	6.25 cd	±	0.04	3.22 e	±	0.03	486.5 fg	±	7.86	1183.5 g	±	9.37	56.95 gh	±	3.74	44.85 ef	±	1.27	28.80 g	±	1.63	32.30 efg	±	1.80	6.20 de	±	0.33
Dur3	5.41 gh	±	0.06	2.68 kl	±	0.03	377.0 l	±	8.08	1097.5 k	±	9.74	54.50 hi	±	1.86	40.60 g	±	1.71	22.65 i	±	1.10	33.30 de	±	1.39	6.60 c	±	0.33
BW1	4.94 k	±	0.00	3.02 fg	±	0.02	491.5 ef	±	8.31	1090.5 k	±	7.90	62.65 fg	±	0.50	38.65 g	±	1.59	32.60 cde	±	1.47	32.40 efg	±	1.31	5.30 f	±	0.33
BW2	5.25 hi	±	0.05	3.03 f	±	0.02	478.0 g	±	7.63	1256.5 c	±	9.37	66.90 def	±	1.54	39.90 g	±	1.47	34.00 bc	±	1.39	30.05 h	±	0.94	6.45 cd	±	0.29
Trit1	5.73 f	±	0.04	2.73 ij	±	0.02	336.5 m	±	7.41	1202.5 f	±	9.74	81.35 b	±	2.57	36.50 h	±	1.55	31.85 e	±	1.59	32.65 ef	±	1.43	7.05 b	±	0.37
Trit2	5.69 f	±	0.04	3.39 c	±	0.03	442.0 i	±	4.94	1238.5 d	±	12.31	92.95 a	±	1.88	23.50 k	±	0.82	26.30 h	±	1.14	23.85 k	±	1.02	3.60 ij	±	0.33
Trit3	5.76 f	±	0.05	2.99 gh	±	0.02	462.5 h	±	10.10	1148.0 i	±	9.55	62.00 fg	±	1.59	28.40 j	±	1.14	32.10 e	±	1.14	34.95 c	±	1.67	4.85 g	±	0.45
Bar1	5.71 f	±	0.02	2.52 m	±	0.03	401.5 k	±	5.61	1171.5 gh	±	8.27	88.6 a	±	1.72	55.70 c	±	1.14	13.40 l	±	0.57	19.30 l	±	1.14	5.55 f	±	0.45
Bar2	5.80 f	±	0.05	2.67 kl	±	0.02	497.5 e	±	6.96	1092.0 k	±	7.72	81.85 b	±	1.19	34.85 hi	±	1.67	13.60 l	±	0.65	31.15 gh	±	1.43	3.80 i	±	0.33
Bar3	5.74 f	±	0.06	3.23 e	±	0.02	455.0 h	±	9.43	1134.0 j	±	8.45	65.00 ef	±	2.12	24.40 k	±	0.49	10.70 m	±	0.73	18.70 l	±	0.98	3.35 j	±	0.29
Bar4	6.07 e	±	0.01	3.33 d	±	0.03	547.0 d	±	9.88	1174.5 g	±	9.37	57.00 gh	±	1.59	45.05 ef	±	0.94	15.75 k	±	0.86	27.25 i	±	1.18	7.35 ab	±	0.37
Em1	6.56 b	±	0.03	2.70 jk	±	0.02	368.5 l	±	7.86	1220.0 e	±	8.08	61.95 fg	±	0.82	59.65 b	±	2.16	34.20 b	±	1.63	35.00 c	±	1.55	7.55 a	±	0.45
Em2	6.41 bc	±	0.07	2.24 n	±	0.02	258.5 n	±	8.31	1039.0 l	±	7.35	68.20 cdef	±	2.65	49.85 f	±	1.92	32.50 de	±	1.39	38.00 b	±	1.22	7.70 a	±	0.41
Spelt	7.94 a	±	0.08	3.76 a	±	0.03	538.0 d	±	7.63	1568.5 a	±	11.57	63.10 fg	±	1.91	57.50 c	±	2.37	39.20 a	±	1.88	70.25 a	±	2.33	7.15 b	±	0.53
Rye1	5.10 ijk	±	0.01	2.67 l	±	0.02	414.0 j	±	6.29	1040.5 l	±	7.53	72.00 cd	±	1.59	62.70 a	±	1.47	23.60 i	±	1.22	31.35 fgh	±	0.94	6.20 de	±	0.24
Rye2	5.08 ijk	±	0.05	2.76 i	±	0.03	464.0 h	±	6.29	1040.5 l	±	7.90	73.85 c	±	1.41	45.55 e	±	1.02	21.05 j	±	1.18	32.55 ef	±	1.59	6.15 de	±	0.45
Oats1	5.47 g	±	0.08	3.02 fg	±	0.02	816.5 a	±	7.41	1161.5 h	±	11.94	50.00 i	±	2.12	34.00 i	±	1.14	33.60 bcd	±	0.73	34.40 cd	±	0.98	4.55 gh	±	0.37
Oats2	5.15 ij	±	0.06	3.48 b	±	0.01	761.5 b	±	7.86	1140.0 ij	±	7.35	42.00 j	±	1.59	43.00 f	±	1.63	26.85 h	±	1.10	25.45 j	±	1.02	4.35 h	±	0.12
Oats3	5.01 jk	±	0.04	2.51 m	±	0.02	729.0 c	±	8.98	1097.0 k	±	9.55	71.00 cde	±	1.06	61.00 ab	±	2.86	33.90 bcd	±	0.82	32.10 efg	±	1.63	6.00 e	±	0.41
Tukey HSD (*p* < 0.05)	0.032			0.006			1.87			2.32			1.17			0.39			0.26			0.25			0.07		

* Average ± standard error; the values signed with the same letter are not significantly different at the 0.05 level of significance.

**Table 3 foods-13-04116-t003:** Phenolic compounds in the grains of examined standard and ancient grains: gallic acid (GA), Cctechin (CAT), dihydrocaffeic acid (DiCA), epicatechin (EPI), dihydro-*p*-coumaric acid (di-p-CoumA), naringin (NA), daidzein (DA), cinnamic acid (CIN), naringenin (NG), chlorogenic acid (ChlA), caffeic acid (CA), p-coumaric acid (p-CoumA), ferulic acid (FA), isoferulic acid (IFA), and quercetin (QUE).

	GA			CAT			EPI			DiCA			di-p-CoumA			NA			NG			DA		
Dur1		n.d.			n.d.			n.d.		74.60	±	0.93	36.77	±	0.54	38.53	±	0.48		n.d.			n.d.	
Dur2		n.d.			n.d.			n.d.		59.03	±	0.85		n.d.		37.04	±	0.53		n.d.			n.d.	
Dur3		n.d.			n.d.		38.51	±	0.06	54.19	±	0.20	43.35	±	0.61	40.92	±	0.51		n.d.			n.d.	
BW1		n.d.			n.d.			n.d.		65.68	±	0.84	45.19	±	0.25	38.05	±	0.22		n.d.		25.48	±	0.34
BW2		n.d.		31.07	±	0.70	38.72	±	0.32	83.28	±	0.56	61.72	±	0.78	40.33	±	0.50	46.42	±	0.58		n.d.	
Trit1		n.d.			n.d.			n.d.		87.32	±	1.10	32.08	±	0.17	39.96	±	0.51		n.d.		26.15	±	0.38
Trit2		n.d.			n.d.			n.d.		113.7	±	1.47	31.79	±	0.17	39.31	±	0.50		n.d.		25.57	±	0.37
Trit3		n.d.			n.d.		39.30	±	0.03	87.92	±	0.72	36.82	±	0.17	38.01	±	0.48		n.d.		25.63	±	0.37
Bar1	33.36	±	5.88 *	443.29	±	5.80	44.61	±	0.12	38.89	±	0.12	48.18	±	0.25	38.31	±	0.21		n.d.		28.54	±	0.37
Bar2	35.13	±	6.09	459.19	±	6.04	41.86	±	0.06	39.81	±	0.04	42.65	±	0.20	38.30	±	0.49	46.66	±	0.56	26.89	±	0.59
Bar3	31.52	±	4.71	360.52	±	4.65	39.13	±	0.10	37.45	±	0.06	42.13	±	0.18	34.02	±	0.45		n.d.		27.68	±	0.40
Bar4	31.16	±	5.57	426.00	±	5.46	45.31	±	0.33	50.07	±	0.22	62.54	±	0.45	36.69	±	0.48		n.d.		27.43	±	0.40
Em1		n.d.			n.d.			n.d.		105.9	±	1.36	30.03	±	0.13	38.52	±	0.50	49.51	±	0.71	28.77	±	0.62
Em2		n.d.			n.d.			n.d.		97.42	±	1.25	28.45	±	0.13	36.24	±	0.47	52.33	±	0.76	26.45	±	0.65
Spelt		n.d.			n.d.		42.34	±	0.25	64.28	±	0.52	22.45	±	0.45	35.22	±	0.51		n.d.			n.d.	
Rye1		n.d.		35.57	±	0.99		n.d.		79.45	±	1.15		n.d.		36.44	±	0.53		n.d.			n.d.	
Rye2		n.d.		33.84	±	1.17		n.d.		92.59	±	1.34		n.d.		37.78	±	0.55		n.d.			n.d.	
Oats1	31.48	±	0.45		n.d.		121.69	±	2.28		n.d.		231.6	±	2.71	50.68	±	0.66	47.10	±	0.64	38.30	±	0.63
Oats2	32.90	±	0.47		n.d.		100.26	±	3.05		n.d.		280.42	±	3.47	48.22	±	0.61	42.35	±	0.64	32.48	±	0.55
Oats3	34.23	±	0.49		n.d.		114.25	±	2.06		n.d.		210.2	±	2.45	45.84	±	0.60	45.22	±	0.61	35.45	±	0.59
	ChlA			CA			*p*-CoumA			FA			CIN			IFA			QUE					
Dur1		n.d.			n.d.		30.59	±	0.03	28.34	±	0.28	21.44	±	0.31	28.74	±	0.35	8.87	±	24.44			
Dur2		n.d.			n.d.		30.81	±	0.03	29.23	±	0.27		n.d.		28.60	±	0.32	10.38	±	21.89			
Dur3		n.d.			n.d.		27.62	±	0.02	29.34	±	0.33	21.55	±	0.31	29.18	±	0.40	6.63	±	15.97			
BW1	27.95	±	0.02	28.58	±	0.05	27.26	±	0.05	30.90	±	0.28	21.48	±	0.37	28.02	±	0.31	10.23	±	12.75			
BW2	27.77	±	0.42		n.d.		29.93	±	0.02	28.72	±	0.31	21.21	±	0.33	28.31	±	0.38	6.83	±	26.98			
Trit1		n.d.			n.d.		29.20	±	0.03	30.18	±	0.33		n.d.		31.40	±	0.42	7.64	±	13.88			
Trit2		n.d.			n.d.		27.60	±	0.03	29.57	±	0.30		n.d.		29.00	±	0.37	8.29	±	19.79			
Trit3		n.d.			n.d.		28.72	±	0.06	30.46	±	0.33		n.d.		33.43	±	0.43	9.39	±	15.60			
Bar1	96.68	±	0.98	29.85	±	0.08	27.44	±	0.45	33.80	±	0.42	21.21	±	1.27		n.d.		14.59	±	19.55			
Bar2	73.30	±	0.63	29.07	±	0.04	29.95	±	0.45	32.26	±	0.40		n.d.			n.d.		15.40	±	31.26			
Bar3	89.14	±	0.87	29.08	±	0.03	28.31	±	0.42	30.39	±	0.38		n.d.			n.d.		14.84	±	16.11			
Bar4	100.6	±	1.13	31.13	±	0.03	32.04	±	0.47	33.13	±	0.42		n.d.			n.d.		13.94	±	37.96			
Em1		n.d.			n.d.		28.45	±	0.42	30.11	±	0.39		n.d.			n.d.		8.47	±	26.17			
Em2		n.d.			n.d.		31.45	±	0.48	34.22	±	0.45		n.d.			n.d.		7.25	±	28.54			
Spelt		n.d.		n.d.			34.55	±	0.09	28.56	±	0.31		n.d.		27.89	±	0.38	6.45	±	30.69			
Rye1		n.d.		n.d.			36.85	±	0.11	30.45	±	0.31		n.d.		28.85	±	0.36	8.45	±	21.86			
Rye2		n.d.		n.d.			32.83	±	0.06	28.35	±	0.29		n.d.		30.53	±	0.37	9.51	±	18.75			
Oats1		n.d.		41.55	±	0.56	34.26	±	0.66	76.28	±	0.68		n.d.		27.48	±	0.07	31.39	±	41.30			
Oats2		n.d.		45.88	±	0.55	36.23	±	0.69	78.42	±	0.76		n.d.		26.56	±	0.02	25.45	±	42.18			
Oats3		n.d.		38.22	±	0.29	42.56	±	0.41	60.22	±	0.47		n.d.		27.45	±	0.02	28.56	±	35.44			

* Average ± standard error; n.d.—not detectable.

**Table 4 foods-13-04116-t004:** DPPH radical scavenging activity and ferric-reducing antioxidant power (FRAP) of extracts in the examined standard grains and ancient grains.

Genotypes	DPPH			FRAP		
	µmol TE·g^−1^			µmol Fe^2+^ eq·g^−1^		
Dur1	201.1 a	±	1.88 *	20.81 m	±	0.29
Dur2	183.7 cd	±	4.95	29.87 g	±	0.47
Dur3	192.8 b	±	7.87	25.03 j	±	0.47
BW1	173.6 e	±	3.75	26.97 i	±	0.41
BW2	155.4 fgh	±	0.79	30.45 fg	±	0.38
Trit1	135.4 k	±	4.77	36.45 de	±	0.55
Trit2	149.3 ghi	±	4.33	23.40 k	±	0.38
Trit3	111.8 l	±	2.92	18.78 n	±	0.28
Bar1	117.8 l	±	5.99	36.12 e	±	0.57
Bar2	141.7 jk	±	3.32	37.39 cd	±	0.52
Bar3	150.2 gh	±	4.76	28.79 h	±	0.45
Bar4	183.5 cd	±	2.24	31.31 f	±	0.42
Em1	177.1 de	±	0.86	22.22 l	±	0.29
Em2	180.5 cde	±	6.36	25.61 j	±	0.45
Spelt	187.3 bc	±	2.50	28.32 h	±	0.40
Rye1	159.4 f	±	4.00	23.09 kl	±	0.36
Rye2	177.7 de	±	6.36	20.01 m	±	0.38
Oats1	142.4 ijk	±	2.13	42.15 b	±	0.56
Oats2	148.2 hij	±	5.49	45.32 a	±	0.67
Oats3	156.1 fg	±	0.63	37.99 c	±	0.48
Tukey HSD (*p* < 0.05)	1.33			0.198		

* Average ± standard error; the values signed with the same letter are not significantly different at the 0.05 level of significance.

## Data Availability

The original contributions presented in the study are included in the article/Appendix A, further inquiries can be directed to the corresponding author.

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
