# Peer review of "Screening of Nutritionally Important Components in Standard and Ancient Cereals"

_foods, 2024, doi:10.3390/foods13244116_

Round 1

Reviewer 1 Report

Comments and Suggestions for Authors

The topic of the manuscript “Screening of nutritionally important components in some standard and ancient small grains” is very relevant and important.

However, there are some serious issues. Firstly and the most importantly is that the presented resulted are full of errors and illogical conclusions that shew doubts on the validity and trustworthiness of the statistical analyses, namely comparison of means and PCA. The dendrograms and PCA are not very informative and their presentation is unclear, some parts of discussion are incorrect, misleading and confusing.

Also, the study lack the detailed information, description and analyses of the environmental conditions that affected the grain quality.

Bellow you may find detailed comments, both more and less important. The less important comments, which the authors will surely recognize, could be taken into consideration only after addressing the most critical ones first.

Abstract

-          The abstract is lacking exact results in terms of values of the most important analyzed parameters.

-          Lines 20-21: The statement that the Cimmyt spelt 1 has high yellow pigment, capacity to reduce DPPH radical, and particularly Zn should be supported with the numerical results in brackets, e.g. „...attention should be given to the Cimmyt spelt 1, which high in yellow pigment (5.01 μg g-1), capacity to reduce DPPH radical (186.2 μmol TE g-1), and particularly Zn (70.25 mg kg-1)...

-          Lines 21-24: The statement ”The presence of phenolics, dihydro-p-coumaric acid, naringin, quercetin, catechin, epicatechin in grains of oats (Sopot) and barley (Apolon and Osvit) underline their unique chemical profile, making them desirable genetic pool for breeding genotypes high in antioxidants and prebiotic compounds.” is not entirely true. In the Table 5, the presence of catechin was not detected in either of the analysed oats.

-           

Introduction

-          Line 48: Kamut is a brand name. Only one of the four cited papers mentions kamut. It is better use in common name khorasan wheat or its latin name.

-          Lines 50-51: Sustainability cannot be a traits. “Aspect” or some other more suitable term should be used instead.

-          Line 86: There are two identical numbers for the cited papers. [11,12,29,29].

Materials and Methods

-          Line 109: A comma is missing betweenbarleyand emmer wheat“.

-          Lines 112-113: What does it exactly mean that “grain from all genotypes originated from 2022”? Please clarify more. Are the grains it collected from one particular field in 2022 where it was replicated? If yes, please give more information about the location and environmental conditions (meteorological and pedological data), the crop management data (fertilizers used, soil type). If not, please explain. The effect of the environment on the variation of nutritional components in the grain is very important, so this information is crucial to set the experimental design correctly and understand the obtained results.

-          Table 1: The barley common type of barley that is not naked barley (or hulless barley) is called hulled barley. Chaff barley is not a standard name.

-          Table 1: Are Sopot – black oats, Caramel oats and brown oats Avena sativa or some other oats species? If they are different species, please add their Latin names in the table.

-          Lines 199-122 and lines 230-234: No need to capitalize the first letters of the compounds.

-          Line 261: LSD was used to test the differences between samples, but LSD is very robust for multiple comparisons as it increases familywise error rate. There are other tests, such as Tukey HSD test, that are more appropriate choice.

-           

Results

-          Table 3. Please check the letter labels. Some are not correct. For instance, in column “Cu” value for Dur2 (6.20f ± 0.33) and BW2 (6.45g ± 0.29) show NOT be significantly different considering that LSD is 0.36. Likewise, Bar4 (7.35h ± 0.37) and Em1 (7.55i ± 0.45) should NOT have different letters. Another one is in the column Mn (Em1 and Oats1).There could be many more errors like these, either here or in other tables. Here it is! Table2, column Pphy, samples Bar 1 and Bar2. And in Table 6, for DHHP, Dur1 and Dur3. Sorry, but I could not check every single one. There is definitely something wrong with either calculations or assigning letters to the values. It is very peculiar that there is not a single values that have more than one letters and that raises some doubts that these results are analyzed and presented correctly. 

-          Lines 301-306: I am not very convinced that the visualization of the relations of the samples with a dendrogram is very informative here. Besides, the interpretation of the dendrogram is not correct. There is a simple proper method to determine the number of clusters. When interpreting the dendrogram, an arbitrary cross-section line can be made at any level, but it is not appropriate to claim the membership to a particular cluster of the samples which branches that cross-section line cuts across. If you choose to have three clusters that they must be: 1) Oats, 2) Em2 and 3) the rests, regardless of a single sample in a cluster. If you choose to draw a line at 99.90% similarity distance, that there are 4 clusters.  If you choose to draw a line at 99.96% similarity distance, that there are 5 clusters.

-          Line 313: “Nevertheless” seems a bit awkward and the sentence would read better without it.

-          Line 314:  The contrast that the word “although” implies is not logical, and it should be replaced with “while” or “whereas” or similar.

-          Lines 381-383: Results showed that all three oats genotypes and Bar3 were mostly determined by β-glucan, arabinoxylan, GSH, Pi, and Ca 382 concentration.

-          Lines 371-379; lines 389-393: It would be helpful to provide a table with loadings (correlation coefficients) between the principal coordinates (PC1, PC2, PC3 and PC4) and the variables (nutritional components) in a supplementary table to back-up the information given in the manuscript.

-          The presentation of the dimensionality and the special positions of phytochemicals and genotypes in the PCA in Fig 4. and Fig 5. are not very clear and obvious, and it lack perspective.  Such obscure presentation may lead to incorrect or illogical interpretation of the PCA, as it is state in lines 396-398: “All three emmer wheat genotypes were determined with concentration of CIN, including also rye, triticale, bread and durum wheat, as well as spelt, but to a lower extent.” It makes very little sense that three emmer genotypes (and to lesser extent rye, triticale and spelt) were determined by the CIN concentration, when they CIN concentrations were NOT detectable, as stated in Table5.

Discussion

-          Lines 435-438: “Regarding significant variability among analysed genotypes in the phytochemical composition of the grains, the results showed that samples of Bar2, triticale, durum, spelt, and emmer wheat had a high concentration of protein, similar to the results obtained by Majzoobi et al. [10].” I would challenge this statement. The protein content below 13% is not considered high, so only Dur1, Bar2, Spelt, Trit3 could be considered to have high protein content. Besides, the values in Table 2 are NOT similar to the results of   Majzoobi et al. who found higher protein content in spelt and emmer.  

-          Lines 439-422: This sentence must be rephrased so to be unambiguous what are the results of his study and what are the results of cited studies.

-          Lines 485-487: ”The results from this study revealed the presence of quercetin, daidzein, naringin, naringenin, catechin, and epicatechin in the tested whole-grain cereals including ancient grains, such as spelt and emmer wheat.” According to the Table 5. daidzein, naringenin and catechin were not detected in the spelt genotype. Neither were epicatechin and catechin detected in emmer. Therefore, this sentence has no sense.

-          Lines 512-520 are mere repetition of the results, and therefore redundant here. The discussion of the PCA is very poor.

Conclusion

- Lines 537-541: “High concentration of phenolic compounds, and particularly the presence of dihydro-p-coumaric acid, naringin, quercetin, catechin, and epicatechin in grains of oats (Sopot) and barley (Apolon and Osvit) underline their unique nutritive profile, making them desirable genetic pool for breeding cereal genotypes high in antioxidants and prebiotic compounds. In the oats genotype Sopot (Oats1) catechin was not detectable!

 Literature

-          The following reference are not complete: 2, 3, 6, 12, 14, 17, 19, 20, 23, 24, 27, 30, 41, 44, 47, 50, 51, 54, 58-61. 

Author Response

The topic of the manuscript “Screening of nutritionally important components in some standard and ancient small grains” is very relevant and important.

However, there are some serious issues. Firstly and the most importantly is that the presented resulted are full of errors and illogical conclusions that shew doubts on the validity and trustworthiness of the statistical analyses, namely comparison of means and PCA. The dendrograms and PCA are not very informative and their presentation is unclear, some parts of discussion are incorrect, misleading and confusing.

Response: Thank you for the detailed review of the manuscript. We considered all comments totally and accordingly made improvements of the manuscript.

Also, the study lack the detailed information, description and analyses of the environmental conditions that affected the grain quality.

Response: The grains used in this study are from the same year, same field, same environment, so it is irrelevant to compare and discuss impact of the environment. We added some information about soil properties in M&M section, but still, we think that only in the case of several environments or years, those effects could be significant.

Bellow you may find detailed comments, both more and less important. The less important comments, which the authors will surely recognize, could be taken into consideration only after addressing the most critical ones first.

Abstract

The abstract is lacking exact results in terms of values of the most important analyzed parameters.

Response: Thank you for your comments. The values were added in the Abstract.

-          Lines 20-21: The statement that the Cimmyt spelt 1 has high yellow pigment, capacity to reduce DPPH radical, and particularly Zn should be supported with the numerical results in brackets, e.g. „...attention should be given to the Cimmyt spelt 1, which high in yellow pigment (5.01 μg g-1), capacity to reduce DPPH radical (186.2 μmol TE g-1), and particularly Zn (70.25 mg kg-1)...“

Response: Values were added.

-          Lines 21-24: The statement ”The presence of phenolics, dihydro-p-coumaric acid, naringin, quercetin, catechin, epicatechin in grains of oats (Sopot) and barley (Apolon and Osvit) underline their unique chemical profile, making them desirable genetic pool for breeding genotypes high in antioxidants and prebiotic compounds.” is not entirely true. In the Table 5, the presence of catechin was not detected in either of the analysed oats.

Response: Higher levels of catchen were detected in barley grains, so the sentence was modified accordingly: The presence of phenolics, dihydro-p-coumaric acid, naringin, quercetin, epicatechin in grains of oats (Sopot), including catechin, in barley grains (Apolon and Osvit)…

-        Introduction

-          Line 48: Kamut is a brand name. Only one of the four cited papers mentions kamut. It is better use in common name khorasan wheat or its latin name.

Response: Thank you, we made change in the text.

-          Lines 50-51: Sustainability cannot be a traits. “Aspect” or some other more suitable term should be used instead.

Response: Thank you, we changed trait to the quality.

         Line 86: There are two identical numbers for the cited papers. [11,12,29,29].

Response: Thank you. We deleted repeated number.

Materials and Methods

-          Line 109: A comma is missing between“barley“ and “emmer wheat“.

Response: Thank you, comma was added.

-          Lines 112-113: What does it exactly mean that “grain from all genotypes originated from 2022”? Please clarify more. Are the grains it collected from one particular field in 2022 where it was replicated? If yes, please give more information about the location and environmental conditions (meteorological and pedological data), the crop management data (fertilizers used, soil type). If not, please explain. The effect of the environment on the variation of nutritional components in the grain is very important, so this information is crucial to set the experimental design correctly and understand the obtained results.

Response: We added required information.

-          Table 1: The barley common type of barley that is not naked barley (or hulless barley) is called hulled barley. Chaff barley is not a standard name.

Response: Thank you. We change it.

-          Table 1: Are Sopot – black oats, Caramel oats and brown oats Avena sativa or some other oats species? If they are different species, please add their Latin names in the table.

Response: They are the same species, as it was marked in the column with species.

-          Lines 199-122 and lines 230-234: No need to capitalize the first letters of the compounds.

Response: We made changes, accordingly.

-          Line 261: LSD was used to test the differences between samples, but LSD is very robust for multiple comparisons as it increases familywise error rate. There are other tests, such as Tukey HSD test, that are more appropriate choice.

Response: Thank you for the notice, but during consulting with statistician working in breeding programs, we were advised to use more robust models, to express variability across genotypes.

   Results

-          Table 3. Please check the letter labels. Some are not correct. For instance, in column “Cu” value for Dur2 (6.20f ± 0.33) and BW2 (6.45g ± 0.29) show NOT be significantly different considering that LSD is 0.36. Likewise, Bar4 (7.35h ± 0.37) and Em1 (7.55i ± 0.45) should NOT have different letters. Another one is in the column Mn (Em1 and Oats1).There could be many more errors like these, either here or in other tables. Here it is! Table2, column Pphy, samples Bar 1 and Bar2. And in Table 6, for DHHP, Dur1 and Dur3. Sorry, but I could not check every single one. There is definitely something wrong with either calculations or assigning letters to the values. It is very peculiar that there is not a single values that have more than one letters and that raises some doubts that these results are analyzed and presented correctly

Response: Thank you for the comment, but actually in Cu column Oats3 have value 6.00f, meaning that 6.00+0.36=6,36, whereas BW2 have value 6.45, which is actually greater than 6.36. Regarding Bar4 and Em1: Trit1 with value 7.05 h and so 7.05+0.36=7.41, thus, Em1 have value 7.55, which is actually greater than 7.41. Also, for Mn column: Oats1 have value 33.60j, meaning that 33.60+1.22=34.82, so Oats1 (33.6), Em1 (34.20), Oats3 (33.9 and BW2(34.00) are all signed with letter j.

Accordingly, we checked all tables with LSD values again.

-          Lines 301-306: I am not very convinced that the visualization of the relations of the samples with a dendrogram is very informative here. Besides, the interpretation of the dendrogram is not correct. There is a simple proper method to determine the number of clusters. When interpreting the dendrogram, an arbitrary cross-section line can be made at any level, but it is not appropriate to claim the membership to a particular cluster of the samples which branches that cross-section line cuts across. If you choose to have three clusters that they must be: 1) Oats, 2) Em2 and 3) the rests, regardless of a single sample in a cluster. If you choose to draw a line at 99.90% similarity distance, that there are 4 clusters.  If you choose to draw a line at 99.96% similarity distance, that there are 5 clusters.

Response: Thank you for the comment. We changed interpretation according to your comments, whereas in the Figure 1 are 3 clusters, in Figure 2 and Figure 3 are two clusters.

-          Line 313: “Nevertheless” seems a bit awkward and the sentence would read better without it.

Response: “Nevertheless” is deleted.

-          Line 314:  The contrast that the word “although” implies is not logical, and it should be replaced with “while” or “whereas” or similar.

Response: Thank you for the valuable comment. Although was changed to whereas.

-          Lines 381-383: Results showed that all three oats genotypes and Bar3 were mostly determined by β-glucan, arabinoxylan, GSH, Pi, and Ca 382 concentration.

Response: Thank you for the notice, sentence was corrected: Results showed that all three oat genotypes and Bar4 were mostly determined by β-glucan, arabinoxylan, GSH, Pi, and Ca concentration.

-          Lines 371-379; lines 389-393: It would be helpful to provide a table with loadings (correlation coefficients) between the principal coordinates (PC1, PC2, PC3 and PC4) and the variables (nutritional components) in a supplementary table to back-up the information given in the manuscript.

Response: We included Supplementary table 3 with correlation coefficients between tested parameters and PC axes.

-          The presentation of the dimensionality and the special positions of phytochemicals and genotypes in the PCA in Fig 4. and Fig 5. are not very clear and obvious, and it lack perspective.  Such obscure presentation may lead to incorrect or illogical interpretation of the PCA, as it is state in lines 396-398: “All three emmer wheat genotypes were determined with concentration of CIN, including also rye, triticale, bread and durum wheat, as well as spelt, but to a lower extent.” It makes very little sense that three emmer genotypes (and to lesser extent rye, triticale and spelt) were determined by the CIN concentration, when they CIN concentrations were NOT detectable, as stated in Table5.

Response: Thank you for the notice, it was a mistake. We changed sentence: Em1, Em2 and BW1 were determined with concentration of CIN, while Rye2, Dur1, Dur2, Trit2, Trit3, BW2, as well as spelt, were determined with concentration of DA. Still, we think that 3D model of PCA presents better results, than for instance 2D model, due to the fact, that 2D model excludes 3rd axes, or much more images must be included into manuscript, what could be much more confusing for potential readers.

Discussion

-          Lines 435-438: “Regarding significant variability among analysed genotypes in the phytochemical composition of the grains, the results showed that samples of Bar2, triticale, durum, spelt, and emmer wheat had a high concentration of protein, similar to the results obtained by Majzoobi et al. [10].” I would challenge this statement. The protein content below 13% is not considered high, so only Dur1, Bar2, Spelt, Trit3 could be considered to have high protein content. Besides, the values in Table 2 are NOT similar to the results of   Majzoobi et al. who found higher protein content in spelt and emmer. 

Response: The sentences were changed accordingly: Regarding significant variability among analysed genotypes in the phytochemical composition of the grains, the results showed that samples of Bar2, Bar4, Trit3, Dur1, and Spelt concentration of protein ≥ 13%. Majzoobi et al. [10] obtained great range in protein content among various grains, including barley, spelt and emmer wheat.

-          Lines 439-422: This sentence must be rephrased so to be unambiguous what are the results of his study and what are the results of cited studies.

Response: We did not properly write sentence, it is not comparison with literature data. Sentence was changed accordingly: Our research revealed that the concentration of β-glucan, as a soluble fibre [4,45], was the highest in the samples of barley (particularly Bar2) and oats genotypes, while the samples of oats (mainly Oats1) and rye genotypes were found to be the best sources of arabinoxylan [2,45].

-          Lines 485-487: ”The results from this study revealed the presence of quercetin, daidzein, naringin, naringenin, catechin, and epicatechin in the tested whole-grain cereals including ancient grains, such as spelt and emmer wheat.” According to the Table 5. daidzein, naringenin and catechin were not detected in the spelt genotype. Neither were epicatechin and catechin detected in emmer. Therefore, this sentence has no sense.

Response: Sentence was modified: The results from this study revealed the presence of quercetin, daidzein, naringin, naringenin, catechin, and epicatech in in the tested whole-grain cereals

-          Lines 512-520 are mere repetition of the results, and therefore redundant here. The discussion of the PCA is very poor.

Response: Thank you for the comment, we modified the whole paragraph: To determine the possible unique nutritive fingerprint of tested whole-grain cereals the principal component analysis was carried out. According to the PCA, samples of barley and oats were separated from all other tested samples due to their unique nutritive profile. The content of total protein, ChlA, GA, DA, CA and CAT separated the barley samples, providing its uniqueness as a source of an important bioactive phenolic compounds. Majzoobi et al. [10] characterised barely as a highly nutritious grain, having great range in protein content, including vitamins and minerals. Special attention should be given to the high β-glucan concentration, including phytosterols and polyphenols. Similarly, the content of EPI, QUE, FA, NA, NG, TPC, Pi, GSH, β-glucan, Arabinoxylan, Ca, and di-p-CoumA induced the separation of oat samples, providing their nutritional importance [10], including antioxidant activity, as well as prebiotic role [30]. According to our best knowledge and through literature review, this is the first report of using PCA as a tool for screening a nutritive profile of barley and oat samples.

Conclusion

Lines 537-541: “High concentration of phenolic compounds, and particularly the presence of dihydro-p-coumaric acid, naringin, quercetin, catechin, and epicatechin in grains of oats (Sopot) and barley (Apolon and Osvit) underline their unique nutritive profile, making them desirable genetic pool for breeding cereal genotypes high in antioxidants and prebiotic compounds. In the oats genotype Sopot (Oats1) catechin was not detectable!

Response: Thank you for the valuable comment. Sentence was corrected, accordingly: The presence of phenolics, dihydro-p-coumaric acid, naringin, quercetin, epicatechin in grains of oats (Sopot), including catechin, in barley grains (Apolon and Osvit) underline their unique nutritive profile, making them desirable genetic pool for breeding cereal genotypes high in antioxidants and prebiotic compounds.

 Literature

The following reference are not complete: 2, 3, 6, 12, 14, 17, 19, 20, 23, 24, 27, 30, 41, 44, 47, 50, 51, 54, 58-61. 

Response: References were generated by the Zotero software, according to the style suggested by the journal, so we are unable to change them.

Reviewer 2 Report

Comments and Suggestions for Authors

Dear authors,

your research is interesting and gives valuable data of cereales nutraceutical potential. There are some comments in the maunscript.

Please check that figure caption is below the figure (not on the other page). The tables are huge, may it will be better to put some data into figures. Try to make the manuscript more "reading" frendly and visualy clear. 

All the best,

All the best

Author Response

Dear authors,

your research is interesting and gives valuable data of cereales nutraceutical potential. There are some comments in the maunscript.

Response: Thank you for the valuable comments; we diligently worked on each of them and improved the manuscript consequently. We provided comments below the suggestions in PDF file, in sticky notes. Please, see the attachement.

Please check that figure caption is below the figure (not on the other page). The tables are huge, may it will be better to put some data into figures. Try to make the manuscript more "reading" frendly and visualy clear. 

Response: Maybe figure captions were moved to the next page, following Figure, possibly to the text formation. Due to the text modifications, some pages, Tables and Figures appear different.

We thought to put some data from the tables into Figures, but actually Figures will take much more space than Tables, and it will be impossible to present all the statistics (significance level and SD) in the Figures. We know that Tables are huge, but they include lot of data (analysed parameters), that are a part of the research.

Reviewer 3 Report

Comments and Suggestions for Authors

The manuscript entitled Screening of nutritionally important components in some standard and ancient small grains presents information related to the protein, minerals, bioactive compounds, and fiber. The manuscript has several issues that the authors must attend to. Also, the language must be revised, there are several typo errors. Below are the comments.

-Line 56. Change nutraceutucal by nutraceutical.

Lines 98-99. What was the rationale for analyzing only protein and minerals? What about carbohydrates?

Line 151. Correct formulas.

Line 160. Revise the correct way to write the molar absorption coefficient.

Methodology section. There are many typographical errors in expressing equations, formulas, exponential, etc. The authors must revise carefully.

Section 2.4.5. What was the rationale for analyzing DPPH and FRAP? Both assays are used to evaluate antiradical activity.

Lines 230-234. What was the rationale for selecting the compounds mentioned in the manuscript?

Line 242. Why did the authors not use standard curves?

So, if the authors did not use standard curves, how they can assure the samples have the compounds mentioned in lines 230-234?

Tables 2, 3, 4 and 6 must be included as supplementary material. Information from tables 3, 4 and 6 are also similar to Figures 1, 2 and 3.

Comments on the Quality of English Language

The language must be revised, there are several typo errors.

Author Response

The manuscript entitled Screening of nutritionally important components in some standard and ancient small grains presents information related to the protein, minerals, bioactive compounds, and fiber. The manuscript has several issues that the authors must attend to. Also, the language must be revised, there are several typo errors. Below are the comments.

Response: Thank you for the valuable comments. We improved the manuscript, consequently.

-Line 56. Change nutraceutucal by nutraceutical.

Response: Thank you for the notice, it was changed.

Lines 98-99. What was the rationale for analyzing only protein and minerals? What about carbohydrates?

Response: It is well-known that cereals are a major source of carbohydrates, so there is a lot of research published on this subject. Nowadays, when plant protein sources are of particular importance and focus of the research, cereals, as staple crops and their protein content are of particular importance. Accordingly, there are breeding programs for increasing the protein level in the cereal grains. Thus, the focus of our research is to provide information on the most important nutrients for humans, to signify whole-grain cereals as nutraceuticals, not to repeat what was already known.

Line 151. Correct formulas.

Response: Corrected

Line 160. Revise the correct way to write the molar absorption coefficient.

Response: Thank you for comment, it was corrected.

Methodology section. There are many typographical errors in expressing equations, formulas, exponential, etc. The authors must revise carefully.

Response: Thank you for comment, we corrected all typing errors.

Section 2.4.5. What was the rationale for analyzing DPPH and FRAP? Both assays are used to evaluate antiradical activity.

Response: Of course, both assays are for the evaluation of the antioxidant activity, but they are quite different: while the DPPH assay records general antioxidant activity (most of the enzymatic and non-enzymatic antioxidants, including ascorbate, glutathione, carotenoids, phytic acid, and many others), FRAP assay is mainly focused on phenolic compounds. So, it is always interesting to compare both assays, particularly when phenolic compounds, as well as other compounds with antioxidant activity, are analysed as part of the research.

Lines 230-234. What was the rationale for selecting the compounds mentioned in the manuscript?

Response: It was already mentioned in the Introduction section. While some of the phenolic compounds, such as ferulic and p-coumaric acids are commonly present in cereal grains, there is a gap in research on flavonoids, isoflavones, flavanones, and flavanols, which are important bioactives in human nutrition. Some of the analysed cereals are rich sources of the mentioned bioactives, while others are not. Thus, this research offers new insights into the composition of cereal grains.

Line 242. Why did the authors not use standard curves?

Response: We corrected sentence accordingly: LabSolutions Shimadzu program was used for instrument control, as well as for data acquisition and analysis. Calibration curves were used for the quantitative calculation of each phenolic compound. The concentration was reported as the mean value of three replications and expressed in μg per g of DW sample.

So, if the authors did not use standard curves, how they can assure the samples have the compounds mentioned in lines 230-234?

Response: We used standard curves, as it was mentioned upper.

Tables 2, 3, 4 and 6 must be included as supplementary material. Information from tables 3, 4 and 6 are also similar to Figures 1, 2 and 3.

Response: Information present in Tables 2, 3 and 6 are of basics for the manuscript, irrespective that they are huge. Nevertheless, we decided to include Tables 1 and 4 in supplementary material, to shorten the manuscript length and avoid burdensome data. HCA analysis was used to compare the level of similarity between tested genotypes regarding all present elements (Figure 1), their ratio with phytic acid in general (Figure 2), as well as the concentration of all present antioxidants (Figure 3). Thus results present in the mentioned tables are quite different to the results present in dendrograms. Yes, there are some similarities, regarding botanically related species/genotypes, but general intent was to present similarities across tested traits, to provide information to potential users on the quality of tested genotypes, as well as to the breeders toward specific traits.

Comments on the Quality of English Language

The language must be revised, there are several typo errors.

Response: Manuscript was revised, for typing errors, as well as English language.

Round 2

Reviewer 1 Report

Comments and Suggestions for Authors

The authors addressed most of the comments, but some of the most important ones are not corrected.

1. The authors neither checked all the tables for errors nor did they read carefully my corrections, but actually chose to ignore my suggestion to check the results.

Table 2. Essential elements in the grain of analyzed in the examined standard and ancient grains

- In column “Cu”, Dur2 is 6.20f ± 0.33, BW2 is 6.45g ± 0.29 and LDS is 0.36. So, 6.45 minus 6.20 is 0.25, which is less than 0.36, therefore Dur2 and BW2 cannot have the same letters. I clearly referred to Dur2, and NOT to Oat3, as the authors suggested in their response.

- Likewise, I pointed out an error when comparing Bar4 (7.35h ± 0.37) and Em1 (7.55i ± 0.45) in column “Cu”. The difference between their means are less than LSD. Why did the authors try to prove that they are right by comparing Trit1 with Em1? The mistake is between the values of Bar4 and Em1.

-  In column “Mn”, Em2 is 32.50i ± 1.39. So, 32.50+1.22 equals 33.77, so Oats1 (33.60j ± 0.73) cannot have a different letter!

- In Table 1, column Pphy, a difference between the mean values of samples Bar 1 (4.46e ± 0.00) and Bar2 (4.50f ± 0.01) is less (0.04) than LSD value (0.074).

-In Table 4, for DHHP, Dur1 (201.9e ± 1.88) and Dur3 (194.2d ± 7.87) have a difference of their means (7.7) less than LSD (18.78).

- The same objection applies for Oats3 - Oats2 and for Oats3 – Oats1 for DPPH in Table 4.

- The same objection applies for Oats1 - Oats2, Oats2 – Oats3, Spelt-Rye1, BW1-Dur3, Trit1-BW2, Spelt-Em1, Spelt-Em2   for FRAP in Table 4.

The interpretation of the statistical results is incorrect! The authors should take this seriously and check all the tables for errors thoroughly. There are just too many mistakes!

2. My comment about the proper method to determine the number of clusters was accepted, but it did not mean that the authors should not comment the sub-clusters using the same method. Why did they reduce the text for each cluster to a single line?! Nobody said so.

3. Regarding my comment on a statement that a genotype cannot be determined by a concentration of a component that was not detectable, the authors changed the sentence to: Em1, Em2 and BW1 were determined with concentration of CIN, while Rye2, Dur1, Dur2, Trit2, Trit3, BW2, as well as spelt, were determined with concentration of DA.

How has a trait determine a genotype when it was not detected in that genotype?! If you look at the table, CIN was not detectable in Em1 and Em2. DA was not detectable in Dur1, Dur2, BW2 and Spelt!

I am not sure whether the authors are in such a hurry to submit and publish or they just did not understand the point.

4. Please add the Latin names of the cereals in the supplement table.

Author Response

The authors addressed most of the comments, but some of the most important ones are not corrected.

Thank you for the understanding and effort to correct and improve the manuscript. We have lesser time to reply to all suggestions; we provided explanation at the end of review. We are particularly grateful for vigilant analysis of present data and statistics.

  1. The authors neither checked all the tables for errors nor did they read carefully my corrections, but actually chose to ignore my suggestion to check the results.

Reply: Thank you for reminding us again. We checked results again, now we compared several statistical programs (including MSTATIC, counting all decimal places to avoid mistakes, while values in tables counted maximum of two decimal places) to correct all issues. Sometimes, after data processing, range determined by software (mean±lsd value) does not match actual range, determined by the simple calculation, so the errors could occur.

Table 2. Essential elements in the grain of analyzed in the examined standard and ancient grains

Reply: Thank you for the notice, we changed accordingly: Essential elements in the grain of the examined standard and ancient grains

- In column “Cu”, Dur2 is 6.20f ± 0.33, BW2 is 6.45g ± 0.29 and LDS is 0.36. So, 6.45 minus 6.20 is 0.25, which is less than 0.36, therefore Dur2 and BW2 cannot have the same letters. I clearly referred to Dur2, and NOT to Oat3, as the authors suggested in their response.

Reply: We corrected all issues and made changes in Tables 1 and 2. We agree that there were a lot of mistakes and that we should be more careful in preparation of tables, as well as results interpretation.

- Likewise, I pointed out an error when comparing Bar4 (7.35h ± 0.37) and Em1 (7.55i ± 0.45) in column “Cu”. The difference between their means are less than LSD. Why did the authors try to prove that they are right by comparing Trit1 with Em1? The mistake is between the values of Bar4 and Em1.

Reply: We corrected all letters following the numbers in table, thus Bar4 is signified as hi, while Em1 is signified as i. Some differences may occur due to the number of decimal places, since we used eight decimal places during data processing.

-  In column “Mn”, Em2 is 32.50i ± 1.39. So, 32.50+1.22 equals 33.77, so Oats1 (33.60j ± 0.73) cannot have a different letter!

Reply: Accordingly, Em2 is now signified as ij, while Oats1 is signified as j.

- In Table 1, column Pphy, a difference between the mean values of samples Bar 1 (4.46e ± 0.00) and Bar2 (4.50f ± 0.01) is less (0.04) than LSD value (0.074).

Reply: Yes, according to the calculation, we corrected and signified Bar1 as d, and Bar2 as de.

-In Table 4, for DHHP, Dur1 (201.9e ± 1.88) and Dur3 (194.2d ± 7.87) have a difference of their means (7.7) less than LSD (18.78).

Reply: Yes, according to the calculation, we corrected and signified Dur1 with letter e, and Dur3 with letter de.

- The same objection applies for Oats3 - Oats2 and for Oats3 – Oats1 for DPPH in Table 4.

Reply: Thank you for the notice; accordingly we signified Oats1 and Oats2 with BC and Oats3 with c letter.

- The same objection applies for Oats1 - Oats2, Oats2 – Oats3, Spelt-Rye1, BW1-Dur3, Trit1-BW2, Spelt-Em1, Spelt-Em2   for FRAP in Table 4.

Reply: Thank you for the notice, we checked again and corrected all letters in the following table.

The interpretation of the statistical results is incorrect! The authors should take this seriously and check all the tables for errors thoroughly. There are just too many mistakes!

Reply: We checked all results again, now we compared several statistical programs to provide correct interpretation of the results.

  1. My comment about the proper method to determine the number of clusters was accepted, but it did not mean that the authors should not comment the sub-clusters using the same method. Why did they reduce the text for each cluster to a single line?! Nobody said so.

Reply: Sorry for the misunderstanding. Text was modified accordingly, for the Figure 1: According to the cluster analysis encompassing all examined elements, following clusters were present (Figure 1): the 1st consisted only Oats2, and the 2nd onewas Em2. The 3rd comprised all other genotypes which consists of a three sub-clusters, where samples of Dur1, Dur3, Trit1, and Spelt created the first sub-cluster, Dur2, Trit3, Rye1, Rye2, Bar, and Bar4 formed the second sub-cluster and BW1, BW2, Trit2, and Em1 were grouped into the third sub-cluster. Bar1 and Bar2, which were independent to genotypes grouped in sub-clusters.

For the Figure2: According to the cluster analysis (Figure 2), two with spelt independent of other genotypes, that formed three sub-clusters. Regarding the 1st sub-cluster samples of Dur1, Dur2, and Em2 formed the first subgroup, the samples of Dur3, Bar3, Rye2, and Bar4 created the second subgroup, while the Em1, Oats2, and Oats3 were grouped in the third subgroup. The second sub-cluster consisted of BW1, BW2, Trit1, Trit2, Trit3, and Oats1 while the samples of Bar1 and Rye1 established the third sub-cluster.

For the Figure 3: The samples Trit1 and Oats3 constituted the 1st cluster among all tested samples. The 2nd cluster encompassed all other genotypes, which were divided into two sub-clusters. The 1st one consisted of samples Dur1 and Dur2. Under the 2nd sub-cluster, two sub-sub-clusters were formed, encompassing Du3, Em2, all barley genotypes, both rye genotypes, as well as Oats1 and Oats2 in the 1st sub-sub-cluster. The 2nd sub-sub-cluster included both bread wheat genotypesTrit2, Trit3, Spelt and Em1.

  1. Regarding my comment on a statement that a genotype cannot be determined by a concentration of a component that was not detectable, the authors changed the sentence to: “Em1, Em2 and BW1 were determined with concentration of CIN, while Rye2, Dur1, Dur2, Trit2, Trit3, BW2, as well as spelt, were determined with concentration of DA.“

How has a trait determine a genotype when it was not detected in that genotype?! If you look at the table, CIN was not detectable in Em1 and Em2. DA was not detectable in Dur1, Dur2, BW2 and Spelt!

Reply: Results regarding PCA were checked again and they were modified accordingly: Results showed that all three oat genotypes and Bar4 were mostly determined by β-glucan, arabinoxylan, as well as oat genotypes were determined by GSH, Pi, and Ca concentration. The samples of Dur2 and Em1 were determined by the reduction capacity of DPPH radical, as well as the concentration of Mn and Cu. The concentration of Pphy was the most important trait for Dur1, Dur3, Em2, and BW1, whereas the concentration of protein and Na determined mainly all three triticale genotypes and durum wheat genotypes.

According to the PCA it was established that all barley genotypes were determined mainly by concentration of CAT and ChlA, and to a lower extent by GA, CA, and DA (Figure 5). Em1, Em2 were determined with concentration of DiCA and BW1 was determined mostly by the CIN concentration, while Trit1, Trit2, and Trit3 were determined with concentration of DA . Furthermore, oats genotypes were determined by concentration of FA, p-CoumA, NG, and to a lower extent with QUE and NA.

I am not sure whether the authors are in such a hurry to submit and publish or they just did not understand the point.

Reply: Actually, when we were notified about finished reviewing process (submitted reports) there were two submitted reviews from another two reviewers and we stared to work on their suggestions and comments. We have seven days to finish manuscript correction and when we logged on (6th day) to the page to submit our answers, there was the third report on the page, which was yours. We lack time to correct manuscript carefully, and we thought that if we do not respond to your suggestion, manuscript will be rejected. Now, we have had enough time to review our paper.

  1. Please add the Latin names of the cereals in the supplement table.

Reply: Latin names were added to the Supplementary table 1.

Reviewer 3 Report

Comments and Suggestions for Authors

The authors have addressed all the pointed issues. I have no further comments 

Author Response

The authors have addressed all the pointed issues. I have no further comments 

Reply: Thank you for the support and effort to improve manuscript.